# The Effect of Two Different Light-Curing Units and Curing Times on Bulk-Fill Restorative Materials

**DOI:** 10.3390/polym14091885

**Published:** 2022-05-05

**Authors:** Gokcen Deniz Bayrak, Elif Yaman-Dosdogru, Senem Selvi-Kuvvetli

**Affiliations:** Department of Pediatric Dentistry, Faculty of Dentistry, University of Yeditepe, Bagdat Cd. No. 238, Istanbul 34728, Turkey; elif.yaman@yeditepe.edu.tr (E.Y.-D.); senem.kuvvetli@yeditepe.edu.tr (S.S.-K.)

**Keywords:** bulk-fill, dental materials, hardness ratio, light-curing, volumetric shrinkage

## Abstract

This study aimed to evaluate the effect of two different light-curing units and curing times on the surface microhardness (SMH), compressive strength (CS), and volumetric shrinkage (VS) of four restorative materials (Filtek^TM^ Z250, Filtek^TM^ Bulk Fill Posterior, Beautifil^®^ Bulk Restorative, ACTIVA^TM^ BioACTIVE). For all tests, each material was divided into two groups depending on the curing unit (Woodpecker LED-E and CarboLED), and each curing unit group was further divided into two subgroups according to curing time (10 s and 20 s). SMH was evaluated using a Vickers hardness tester, CS was tested using a universal testing machine, and VS was measured using video imaging. In all the restorative materials cured with Woodpecker LED-E, the 20 s subgroup demonstrated significantly higher SMH values than the 10 s subgroup. In both light-curing time subgroups, the CarboLED group showed significantly higher CS values than the Woodpecker LED-E group for all restorative materials except Filtek^TM^ Bulk Fill Posterior cured for 20 s. ACTIVA^TM^ BioACTIVE showed significantly greater volumetric change than the other restorative materials. A higher curing light intensity and longer curing time had a positive effect on the SMH and CS of the restorative materials tested in this study. On the other hand, curing unit and time did not show a significant effect on the VS values of restorative materials.

## 1. Introduction

Resin-based composites have been extensively used in dentistry because of their improved composition, good aesthetic qualities, and easy handling. However, when restoring cavities with these materials, incremental application with a maximum of 2 mm thickness should be implemented to minimize polymerization shrinkage, microleakage, and postoperative sensitivity [1]. Several manufacturers have introduced bulk-fill restorative materials that can be applied in a single layer up to 4–5 mm thick to reduce the number of clinical steps, the risk of contamination, and the formation of air bubbles [2]. Recently, a novel bioactive bulk-fill restorative material (ACTIVA^TM^ BioACTIVE restorative material) that mimics natural teeth’s physical and chemical properties has been introduced [3].

There are several light-curing devices on the market, including quartz-tungsten halogen (QHT), plasma-arc, laser, and light-emitting diode (LED) versions. Recently, newer generations of LED light-curing devices have been introduced, with a higher range of intensity, to improve factors affecting the clinical performance of restoratives [4]. Moreover, it has been claimed that curing time can be decreased when the irradiance output (mW/cm^2^) is increased, which may be important for pediatric dentistry [5,6].

Dental anxiety and fear are significant challenges for pediatric patients. When these patients are uncooperative, it may become challenging to prevent saliva contamination, which is crucial for the success of the restoration [5,7]. Dental treatments should be as quick and practical as possible for such patients. In cases of deep restoration, several layers should be applied, but this is complex and time-consuming. To eliminate this problem, new generation dental materials (bulk-fill restorative materials) can be used, which shorten the procedure time [5,8]. However, dentists are still distrustful of adopting this new type of material in clinical practice [9]. There are recommended polymerization instructions for each material, but these may not be always applicable in clinical practice. This study will provide insight into how some mechanical and physical properties of the different types of bulk-fill material are affected by a high or low-intensity light device when the polymerization time is shortened.

To our knowledge, there are no studies comparing the surface microhardness (SMH), compressive strength (CS) and volumetric shrinkage (VS) of micro-hybrid composite resin (Filtek^TM^ Z250; 3M ESPE), bulk-fill composite resin (Filtek^TM^ Bulk Fill Posterior; 3M ESPE), giomer-based bulk-fill composite resin (Beautifil^®^ Bulk Restorative; Shofu) and bioactive bulk-fill restorative material (ACTIVA^TM^ BioACTIVE; Pulpdent Corporation) when polymerized with two different light-curing devices (Woodpecker LED-E and CarboLED) for 10 s and 20 s. Therefore, this in vitro study aimed to compare the effects of two different light-curing units and different times on the SMH, CS, and VS of four different restorative materials. The first null hypothesis tested was that the curing units and times would not influence the SMH and CS of restorative materials. The second null hypothesis of this study was that the VS of restorative materials would not be affected by light-curing units and curing times.

## 2. Materials and Methods

A micro-hybrid composite resin [(Filtek^TM^ Z250; 3M ESPE, St. Paul, MN, USA)], a bulk-fill composite resin [(Filtek^TM^ Bulk Fill Posterior; 3M ESPE, St. Paul, MN, USA)], a giomer-based bulk-fill composite resin [(Beautifil^®^ Bulk Restorative; Shofu Inc., Kyoto, Japan), and a bioactive bulk-fill restorative material [(ACTIVA^TM^ BioACTIVE; Pulpdent Corporation, Watertown, MA, USA)] were used in the study. A light shade was selected for each material (shade A1, except for ACTIVA^TM^ BioACTIVE) for optimal light penetration. The curing light units used in this study were Woodpecker LED-E [Woodpecker Medical Instrument Co., Guilin, China] and CarboLED CL-01 [GCP Dental, Ridderkerk, Netherlands]. The power intensity of the light-curing units used was measured using a digital radiometer (Woodpecker LED Light Meter; Woodpecker Medical Instrument Co., Guilin, China). Details of the materials and the light-curing units are listed in Table 1 and Table 2, respectively.

### 2.1. Specimen Preparation

A total of 112 disc-shaped specimens (28 samples from each restorative material) with 6-mm in diameter and 4-mm thickness were prepared for the SMH test. A total of 160 specimens (40 samples from each restorative material) with 4-mm in diameter and 6-mm in thickness were prepared for the CS test. Specimens for each test were divided into two groups depending upon the curing system [Woodpecker LED-E (W) and CarboLED (C)], and each curing unit group was further divided into two subgroups according to light-curing times (10 s and 20 s). 

Customized sectional plexiglass molds with markings at 2 mm depths were used to prepare the samples. The schematic representation of the study design is shown in Figure 1. The molds were placed on a glass slide covered with a Mylar strip. For the SMH test, each material except for the conventional composite resin (Filtek^TM^ Z250) was inserted into the mold in a 4 mm single increment. Composite resin was packed into the mold in 2 increments of 2 mm. For the CS test, similar to a previous study, the composite was placed in 2 mm increments up to 6 mm, and the other materials were inserted in 2 increments of 3 mm each [10]. All materials were polymerized after each increment.

Each layer was photo-polymerized, and the top side of the mold was covered with a Mylar strip adpressed with a glass slide to expel excess material and produce a smooth surface. After removing the glass slide, specimens were polymerized with one of the selected curing units in contact with the top surface of the Mylar strip. The Woodpecker LED-E with an output of 850 mW/cm^2^ and the CarboLED with a light intensity of 1400 mW/cm^2^ were each applied for both 10 s and 20 s. The resin-modified glass ionomer cement (GIC) was allowed to set at room temperature for 5 min. After light-curing and setting, the specimens were removed from the mold, and the top surfaces of the specimens were identified with an indelible mark. The samples were kept in distilled water in complete darkness at 37 °C for 24 h.

### 2.2. Microhardness Measurement

Before testing SMH, both surfaces of each sample were polished with 600-grit silicon carbide paper. The SMH of the top and bottom surfaces was measured using a Vickers micro-hardness tester (Micromet 5114; Buehler, Lake Bluff, IL, USA) under a load of 100 g and dwell time of 15 s. Three readings were recorded for each surface (top and bottom), and results were averaged to obtain a single value. The obtained value was recorded as VHN, and the hardness ratio (VHN of bottom/top) was calculated.

### 2.3. Compressive Strength Measurement

The CS of all specimens was performed using a universal testing machine (Instron 3345, Instron Corp., Norwood, MA, USA) at a crosshead speed of 1 mm/min. CS values were determined in megapascals (MPa) by dividing the failure load (N) by the specimen cross-section area (mm^2^).

### 2.4. Volumetric Polymerization Shrinkage Evaluation

Volumetric shrinkage (n = 7) of four different restorative materials was measured using a video imaging device (Acuvol; Bisco Inc., Schaumburg, IL, USA) in a single-view mode. Each sample of approximately 10 mL was shaped into a hemisphere and placed on the pedestal of the device. The samples were left untouched for 3 min to take their final shape, then the first volume (V_1_) was recorded. Subsequently, the specimens were irradiated using the same light-curing units for 10 and 20 s as in the SMH and CS tests. After 10 min, the post-curing volume (V_2_) was measured, and the final shrinkage value was calculated as follows: VS% = [(V_1_ − V_2_)/V_1_] × 100.

### 2.5. Statistical Analysis

Statistical analysis was performed using the NCSS (Number Cruncher Statistical System, Kaysville, UT, USA) program. The normality of the distributions was confirmed by the Shapiro-Wilk test. One-way analysis of variance (ANOVA) testing was used for intragroup comparisons and the Tukey multiple comparison test was used to determine intergroup differences. Results were evaluated at a level of *p* < 0.05 significance.

## 3. Results

### 3.1. Microhardness

The mean and standard deviation (SD) of SMH values for both surfaces (top and bottom) of all materials are shown in Table 3. The top surface of Filtek Z250 cured with the C light-curing unit for 20 s showed the highest mean SMH value (119.25 ± 0.81). The bottom surface of ACTIVA Bioactive Restorative cured with the W light-curing unit for 10 s recorded the lowest mean SMH value (48.51 ± 1.71). For all materials cured with the W light-curing device, the top and bottom surfaces of the specimens cured for 20 s demonstrated statistically significant higher SMH values than those cured for 10 s (*p* < 0.05). Group C showed statistically significant higher VHN values compared to group W for the top surfaces of all material groups cured for each curing time subgroup, with the exception of the Filtek Bulk Fill cured with both curing units for 20 s (*p* = 0.0001). According to the Tukey multiple comparison test, bioactive bulk-fill restorative material showed significantly lower SMH values on the top surface than the other materials tested for each curing time subgroup of the W and C groups (*p* = 0.0001).

### 3.2. Hardness Ratio

Figure 2 represents the distribution of the mean hardness ratio (VHN of bottom/top) values and SD of the tested materials for all polymerization protocols. The highest mean hardness ratio (VHN values of bottom/top) was recorded for Filtek Z250 composite resin when cured with the C curing system for 10 s (0.91 ± 0.03), while the lowest value was obtained for ACTIVA Bioactive Restorative when cured with the C curing system for 10 s (0.74 ± 0.01). In the 20 s subgroup of all restorative materials except for Filtek Z250, group C produced significantly lower hardness ratio values compared to group W (*p* < 0.05). When the Filtek Bulk Fill material was cured with the W or C light-curing unit, the hardness ratio of the samples polymerized for 20 s was found to be statistically significantly higher than those polymerized for 10 s (*p* = 0.0001). Except for the Filtek Z250, the hardness ratio of all tested materials cured with the C light-curing device for 10 s remained below the acceptable minimum level (0.8), as indicated.

### 3.3. Compressive Strength

The mean CS values for all tested materials are shown in Table 4. Filtek Z250 polymerized using the C light-curing unit for 20 s had the highest mean CS values (245.38 ± 39.09), whereas Filtek Bulk Fill cured with the W light-curing unit for 10 s showed the lowest mean CS values (118.22 ± 14.51). There were statistically significant differences in CS between the W and C groups of all the restorative materials for each curing time, with the exception of the 20 s subgroup of the Filtek Bulk Fill (*p* < 0.05).

### 3.4. Volumetric Shrinkage

Table 5 presents the mean VS values for all materials tested. ACTIVA Bioactive Restorative demonstrated significantly higher VS compared to the other materials tested in all polymerization procedures (*p* = 0.0001). For each light-curing unit group, a longer curing time yielded significantly greater VS than a shorter curing time only for the Beautifil Bulk Fill material (*p* < 0.05). No significant difference was found in VS between the curing time subgroups of all other materials tested (*p* > 0.05).

## 4. Discussion

It is well known that the physical and mechanical properties of resin-based dental materials are dependent on certain variables including light-curing unit, light intensity, wavelength and curing time [5,11,12]. Previous studies stated that higher power density increased the microhardness of resin-based dental materials [13,14]. In this study, all materials polymerized using the C light-curing device demonstrated higher VHN values at the top surface compared to those cured with the W light-curing device in both curing time subgroups, except for the Filtek Bulk Restorative cured for 20 s. This can be explained by the high total energy produced by the CarboLED device and the high energy transferred to the material. Park et al. [15] claimed that curing with wide wavelength width improved the microhardness of materials. Therefore, in this study, the higher SMH values may be due to the wider wavelength of the CarboLED device. Moreover, higher curing time significantly enhanced the SMH values at the top surface in all the materials except for the Filtek Bulk Fill and Beautifil Bulk Restorative cured with C, which was in agreement with previous studies [11,12,16]. Considering the obtained data in this study, it can be inferred that high-power light intensity and extended polymerization time resulted in greater VHN in almost all materials (*p* < 0.05).

For each curing time subgroup, Filtek Z250 cured with the C light-curing unit showed significantly greater VHN values on the top surface than Filtek Bulk Fill and Beautifil Bulk Restorative materials cured with the C light-curing unit. However, there was no significant difference in terms of SMH between the Filtek Bulk Fill and Beautifil Bulk Restorative when cured with C for both 10 and 20 s. This result is in accordance with the findings of previous studies where conventional composite resin materials had higher SMH values compared with bulk-fill resin composite materials [17,18]. It has been claimed that highly filled materials exhibit better mechanical properties [17,19]. In the current study, however, Filtek Z250 (filler content; wt%:84.5%) showed higher SMH values than Beautifil Bulk Restorative (filler content; wt%:87%), which has a higher filler content. This may be explained by the existence of zirconia particles, which can increase the resistance of Filtek Z250 material. ACTIVA Bioactive Restorative showed a significantly lower microhardness value compared to the other materials in all polymerization processes. This may be because ACTIVA restorative material is a glass ionomer-based material, unlike the other materials tested.

The hardness ratio represents the degree of conversion of the deeper surface in relation to the top surface [20]. It is calculated by dividing the VHN of the bottom surface by the VHN of the top surface and it is proposed that the ratio should be at least 0.80 for adequate depth of cure [21]. In the current study, for each curing device and curing time, the hardness ratio of Filtek Z250 was above this value. When the materials were cured with the C light-curing device for 10 s, the hardness ratio of all the materials except for Filtek Z250 remained below this value. Considering this data, polymerization with a high-intensity light device for a short time did not yield favorable results in terms of hardness ratio. Nevertheless, the ratio varied depending on the material, light-curing unit, and curing time. Peutzfeldt and Asmussen [22] stated that the degree of cure decreased with increasing light intensity. Similarly, in this study, when the specimens were cured for 20 s, group C showed a lower hardness ratio compared with group W for each restorative material except for Filtek Z250. It has been asserted that excessive intensity of light can cause rapid conversion, providing immediate polymerization [23,24]. Therefore, the hardness ratio of materials polymerized with high light intensity may be low. A study by Illie et al. [25] found that a short curing time was not enough to provide polymerization on the deeper side of the composite material. In the current study, for the Filtek Bulk Fill and Beautifil Bulk Restorative materials, the 20 s subgroup exhibited a higher hardness ratio compared to the 10 s subgroup. In other materials, the hardness ratio did not increase as the polymerization time increased. This might be attributed to the translucent fillers and the matrix of bulk-fill restorative materials allowing light transmission through the material.

The present study revealed that the hardness ratio of Filtek Z250 restorative material was higher than that of the other materials tested. Garcia et al. [20] reported that the bottom hardness value of materials decreased as the thickness of the samples increased. In the current study, Filtek Z250 was polymerized at 2-mm thickness while the other materials were polymerized at 4-mm thickness. This may have contributed to the comparatively higher hardness ratio of the Filtek Z250.

According to a previous study, higher light intensity output did not significantly enhance the CS values of resin-based restorative materials [26]. Conversely, in other studies, resin-based materials with higher CS were found to be obtained when polymerized with a high-intensity light device. In line with the findings of the previous studies, the CS values of the materials cured with the C light-curing device were significantly higher than those cured with the W light-curing device in the current study [27,28], except for Filtek Bulk Fill cured for 20 s. It has been stated that the CS of resin-based restorative materials did not improve with increasing polymerization time [11]. However, in this study, except for Filtek Bulk Fill cured with C, the CS values of the materials polymerized for 20 s were significantly higher than those polymerized for 10 s. Therefore, the first null hypothesis of this study was rejected.

A previous study showed that the amount of filler load did not influence the physical properties of composite resin materials [29]. However, Baek et al. [26] reported that although there is a correlation between filler load and the CS of restorative materials, this may vary according to the material. In the present study, a relationship was found between the filler load and the CS values of all materials tested, except for ACTIVA Bioactive Restorative. Even though ACTIVA contains less filler, it exhibited similar CS value to Filtek Bulk Fill which has a higher amount of filler. According to the manufacturer, this bioactive material includes a flexible resin matrix and silica glass particles, and can absorb stress. Thus, it can display improved physical and mechanical properties [30]. Therefore, the interpretation of this outcome is that CS values may change depending on the content of the material.

Rizzante et al. [17] observed that there was a strong correlation between filler content and polymerization shrinkage, and shrinkage decreased as filler content increased. However, in this study, Filtek Z250 showed more polymerization shrinkage than Filtek Bulk Fill although the filler load of Filtek Z250 was higher. This may be related to the filler ingredient of Filtek Z250. On the other hand, ACTIVA showed significantly more volumetric change than the other tested materials. In a previous study using video imaging, dual-cure composite cement showed volumetric shrinkage similar to the ACTIVA material tested in this study [31]. This may be due to the filler ratio, as well as the consistency, chemical properties, and the setting mechanism of the material. In addition, Suiter et al. [32] found that the volumetric change of resin-modified GIC after polymerization was greater than compomer and resin composite. However, during water storage, shrinkage was compensated in RMGIC compared to other materials. Because resin-modified GIC is a hydrophilic material, it can exhibit hygroscopic expansion in a humid environment. Therefore, the volumetric change of this material needs to be evaluated after storage in water. Shibasaki et al. [33] reported that bulk-fill resin composites demonstrated higher volumetric change than resin composite materials. Conversely, in other studies, bulk-fill composites showed similar or lower polymerization shrinkage compared to resin composites [17,34]. Furthermore, a recent study by Yu et al. [35] showed that giomer-based bulk-fill composite resin had lower polymerization shrinkage than resin composite. Thus, previous studies support the results observed in the present study, having shown that bulk-fill composites exhibit lower polymerization shrinkage compared to conventional composite resin [17,34,35].

In the present study, for both curing time subgroups, no significant differences in VS were found between the W and C groups of all materials, except for the Filtek Bulk Fill 20 s subgroup. It has been reported in studies that higher energy density and longer curing time yielded greater polymerization shrinkage [36], and that shorter curing time reduced polymerization shrinkage but after 24 h all irradiation times showed similar polymerization shrinkage values [37]. However, Zorzin et al. [38] reported that extended curing time did not significantly affect VS, and some materials even showed less shrinkage after longer polymerization. In another study, it was deduced that no significant increase in VS was observed with an increase in curing time [39]. Similarly, in this study, for both light-curing unit groups, there were no significant differences in VS between the 10 s and 20 s subgroups for each restorative material except Beautifil Bulk Restorative. Based on the results of the present study, the second null hypothesis was accepted.

According to previous studies, it was stated that the depth of cure and the polymerization shrinkage of the Filtek Bulk Fill was higher than that of the Beautifil Bulk Restorative, and this could be due to the fact that the Filtek Bulk Fill was more translucent compared to the Beautifil Bulk Restorative [37,40]. In the current study, the 20 s subgroup demonstrated significantly higher polymerization shrinkage than the 10 s subgroup only in the case of the Beautifil Bulk Restorative. Filler type may influence light-scattering behavior and polymerization shrinkage [41]. Filtek Bulk Fill includes nanocluster fillers, and Beautifil Bulk Restorative contains surface pre-reacted glass-ionomer (S-PRG) fillers. Therefore, the increase in VS of the Beautifil Bulk Restorative when polymerized for longer may be due to its ingredients including S-PRG fillers.

One of the main reasons for the failure of resin-based restorative materials is microleakage resulting from polymerization shrinkage. For this reason, proper curing methods, incremental layering techniques and appropriate material selection are required to reduce polymerization shrinkage [42]. However, the application of these procedures in teeth with excessive tooth tissue loss is not always sufficient for restoration success. In such cases, indirect restorations are needed to eliminate shrinkage. Although the use of indirect restorations in pediatric dentistry is limited, they are generally preferred in the restoration of permanent teeth with excessive dental tissue loss [43].

Recently, with the development of technology, computer-aided design/computer-aided manufacturing (CAD/CAM) and three-dimensional (3D) printing applications have gained popularity in dental treatments. Such applications are preferred in indirect restorations such as inlays, onlays and overlays, apart from prosthetic procedures. Although these techniques are costly compared to conventional techniques, they provide decreased chair-time, good aesthetic quality and marginal precision. They also increase the durability of restorations by eliminating operator and technical errors [44,45]. However, these new technologies require further in vitro and in vivo studies.

To find the most suitable material and light device that gives the best results in the shortest curing time, especially in pediatric dentistry, further studies should be carried out by using other improved light devices and various restorative materials. Also, restoratives should be evaluated in terms of other physicomechanical parameters such as shrinkage stress, microleakage, and tensile strength.

## 5. Conclusions

According to the results of this in vitro study, the SMH and CS of the tested restorative materials improved with an increase in the light-curing intensity and time. The VS of all the tested restorative materials was not significantly influenced by the curing time, except for Beautifil Bulk Restorative. When the tested materials were polymerized for 10 s with a high-intensity light device, superior SMH and CS values were found in Filtek Z250, but the material showed high polymerization shrinkage. For this reason, in patients with poor cooperation, bulk-fill materials can be applied for 10 s with a high-intensity light device when necessary. However, although the bioactive bulk-fill material ACTIVA Bioactive Restorative showed good CS values, it should be used carefully on load-bearing areas due to its low SMH values.

## Figures and Tables

**Figure 1 polymers-14-01885-f001:**
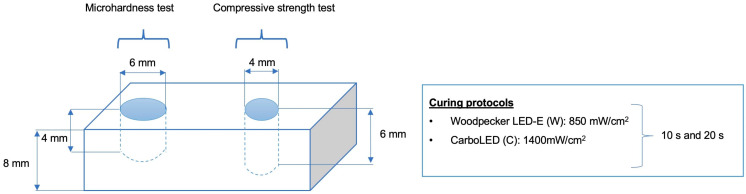
Schematic view of the study design.

**Figure 2 polymers-14-01885-f002:**
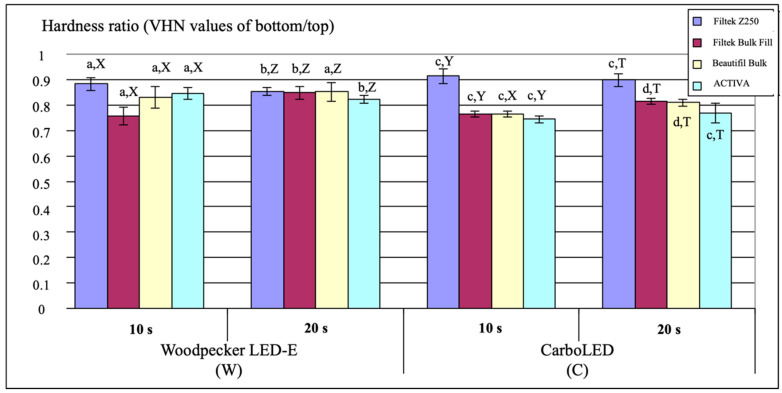
The distribution of the mean VHN values of the bottom/top ratio for each restorative material. The same letters for each restorative material cured with Woodpecker LED-E or CarboLED indicate no significant difference between the hardness ratios of the samples cured for 10 and 20 s. The same uppercase letters for each restorative material cured for 10 s or 20 s indicate no significant difference between the hardness ratios of the samples cured with Woodpecker LED-E and CarboLED (*p* < 0.05).

**Table 1 polymers-14-01885-t001:** Properties of the Tested Restorative Materials as Provided by the Manufacturer.

Material	Shade	Composition	Filler Load wt% (vol%)	Recommended Curing Time and Light Intensity	Recommended Thickness	Manufacturer Lot No.
Filtek^TM^ Z250	A1	Filler:Zirconia/silica Resin matrix: Bis-GMA, UDMA, Bis-EMA, TEGDMA	84.5% (60%)	20 s ≥400 mW/cm^2^	2.5 mm	3M ESPE, St. Paul, MN, USA (N795944)
Filtek^TM^ Bulk Fill	A1	Filler:Zirconia/Silica Ytterbiyum trifloride Resin matrix: UDMA, Bis-GMA, Bis-EMA	76.5% (58.4%)	20 s ≥1000 mW/cm^2^	4 mm	3M ESPE, St. Paul, MN, USA (NA50988)
				40 s 550–1000 mW/cm^2^		
Beautifil^®^ Bulk Restorative	A1	Filler: S-PRG filler based on fluoroboroaluminosilicate glass, polymerization initiator, pigments and others Resin matrix: Bis-GMA, UDMA, Bis-MPEPP, TEGDMA	87% (74.5%)	10 s ≥1000 mW/cm^2^	4 mm	Shofu Co, Kyoto Japan (031828)
ACTIVA™ Bioactive-Restorative	A2	Filler: Modified polyacrylic acid (44.6%), amorphous silica (6.7%), and sodium fluoride (0.75%) Resin matrix: Blend of diurethane and other methacrylates	55.4% (44.6%)	20 s 550–1000 mW/cm^2^	4 mm	Pulpdent, Watertown, USA (190110)

Bis-GMA—bisphenol A glycol dimethacrylate, UDMA—urethane dimethacrylate, Bis-EMA—bisphenol A ethoxylate dimethacrylate, TEGDMA—triethylene glycol dimethacrylate, Bis-MPEPP—bisphenol A polyethoxy methacrylate, S-PRG-surface pre-reacted glass-ionomer.

**Table 2 polymers-14-01885-t002:** The Light-Curing Units Used in the Study.

Light-Curing Unit	Company	Wavelength (nm)	Irradiance (mW/cm^2^)	Serial No.
Woodpecker LED-E (W)	Woodpecker Medical Instrument Co., Guilin, China	420–480	850–1000	L1980545XE
CarboLED (C)	GCP Dental, Ridderkerk, Netherlands	395–480	1400	DYL31406034

**Table 3 polymers-14-01885-t003:** The Mean and SD of Microhardness (VHN) Values for Top and Bottom Surfaces of the Restorative Materials, according to the Different Light-Curing Devices and Curing Times.

Material (n = 7)	Light-Curing Unit	Top Surface (Mean ± SD)		Bottom Surface (Mean ± SD)	
		Curing Time			
		10 s	20 s	10 s	20 s
Filtek Z250	W	105.5 ± 2.21 ^a,A,x^	113.35 ± 1.22 ^a,B,x^	93.2 ± 3.43 ^c,C^	96.81 ± 1.59 ^c,D^
	C	114.13 ± 0.83 ^b,A,X^	119.25 ± 0.81 ^b,B,X^	104.34 ± 2.72 ^d,C^	107.13 ± 2.67 ^d,C^
Filtek Bulk Fill	W	106.98 ± 2.07 ^a,A,x^	113.4 ± 1.71 ^a,B,x^	81 ± 3.17 ^c,C^	96.22 ± 3.87 ^c,D^
	C	112.54 ± 0.64 ^bA,Y^	113.77 ± 1.57 ^a,A,Y^	86.19 ± 1.52 ^d,C^	92.77 ± 1.47 ^d,D^
Beautifil Bulk Restorative	W	103.16 ± 1.8 ^a,A,x^	105.46 ± 0.72 ^a,B,y^	85.62 ± 3.27 ^c,C^	89.91 ± 4.0 ^c,D^
	C	112.75 ± 0.74 ^b,A,Y^	113.39 ± 1.54 ^b,A,Y^	86.21 ± 1.63 ^c,C^	91.92 ± 2.38 ^c,D^
ACTIVA Bioactive Restorative	W	57.22 ± 1.06 ^a,A,y^	63.49 ± 0.99 ^a,B,z^	48.51 ± 1.71 ^c,C^	52.2 ± 0.93 ^c,D^
	C	70.63 ± 1.21 ^b,A,Z^	72.57 ± 1.52 ^b,B,Z^	52.56 ± 0.38 ^d,C^	55.81 ± 2.2 ^d,D^

-Same lowercase first letters within columns indicate no significant difference for individual material at each surface measurement, -Same uppercase second letters within rows indicate no significant difference for each surface measurement, -For the top surfaces, same lowercase third letter within columns denotes no significant difference among W light-curing unit groups, -For the top surfaces, same uppercase third letter within columns denotes no significant difference among C light-curing unit groups (*p* < 0.05), -SD, Standard deviation.

**Table 4 polymers-14-01885-t004:** The Mean and SD of Compressive Strength (MPa) for each Restorative Material.

Material	Light-Curing Unit	Curing Time	
		10 s	20 s
Filtek Z250	W	173.13 ± 19.76 ^a,A,x^	183.63 ± 18.39 ^a,A,x^
	C	201.89 ± 24.43 ^b,A,X^	245.38 ± 39.09 ^b,B,X^
Filtek Bulk Fill	W	118.22 ± 14.51 ^a,A,y^	152.06 ± 12.95 ^a,B,y^
	C	164.91 ± 33.03 ^b,A,Y^	167.29 ± 25.15 ^a,A,Y^
Beautifil Bulk Restorative	W	178.39 ± 9.59 ^a,A,x^	188.97 ± 10.3 ^a,B,x^
	C	188.22 ± 10.85 ^b,A,X,Y^	208.2 ± 11.84 ^b,B,Z^
ACTIVA Bioactive Restorative	W	137.08 ± 13.72 ^a,A,z^	163.03 ± 6.98 ^a,B,y^
	C	166.25 ± 7.95 ^b,A,Y^	179.67 ± 6.94 ^b,B,Y,Z^

-Same lowercase first letter within columns indicates no significant difference for individual materials, -Same uppercase second letter within rows indicates no significant difference for each light-curing unit of each restorative material, -Same lowercase third letter within columns shows no significant difference among W light-curing unit groups, -Same uppercase third and forth letters within columns show no significant difference among C light-curing unit groups (*p* < 0.05). -SD, Standard deviation.

**Table 5 polymers-14-01885-t005:** The Mean and SD of Volumetric Shrinkage Measurements (%) Determined by Video-Imaging.

Material	Light-Curing Unit	Curing Time	
		10 s	20 s
Filtek Z250	W	2.31 ± 0.11 ^a,A,x^	2.34 ± 0.13 ^a,A,x^
	C	2.45 ± 0.40 ^a,A,X^	2.27 ± 0.31 ^a,A,X^
Filtek Bulk Fill	W	1.81 ± 0.22 ^a,A,y^	1.93 ± 0.19 ^a,A,x^
	C	1.86 ± 0.20 ^a,A,Y^	1.64 ± 0.28 ^b,A,Y^
Beautifil Bulk Restorative	W	1.64 ± 0.09 ^a,A,y^	1.78 ± 0.11 ^a,B,x,y^
	C	1.61 ± 0.18 ^a,A,Y^	1.85 ± 0.07 ^a,B,X,Y^
ACTIVA Bioactive Restorative	W	3.70 ± 0.40 ^a,A,z^	3.82 ± 0.54 ^a,A,z^
	C	3.71 ± 0.46 ^a,A,Z^	3.92 ± 0.65 ^a,A,Z^

-Same lowercase first letter within columns indicates no significant difference for each material, -Same uppercase second letter within rows indicates no significant difference for each light-curing unit of each restorative material, -Same lowercase third and forth letters within columns show no significant difference among W light-curing unit groups, -Same uppercase third and forth letters within columns show no significant difference among C light-curing unit groups (*p* < 0.05), -SD, Standard deviation.

## Data Availability

The data presented in this study are contained within the article.

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
