# Peer review of "The Effect of Two Different Light-Curing Units and Curing Times on Bulk-Fill Restorative Materials"

_polymers, 2022, doi:10.3390/polym14091885_

Round 1
Reviewer 1 Report
Interesting paper with good English. I like the way you wrote your materials and methods.
However need some corrections as mention below:
- Abstract is vague in terms of materials used. The author mentioned that ". For all tests, each material was divided into two groups depending on the curing unit, and each curing unit group was further divided into two subgroups according to curing time. Abstract clearly did not say what is the materials and what is each group. This is not acceptable and need correction.
- Section 3.2 you did not provide units in text for instance look at the following sentence: "The highest mean hardness ratio was recorded
for Filtek Z250 composite resin when cured with C curing system for 10 s (0.914 ± 0.028), while the lowest value was obtained for ACTIVA Bioactive Restorative when cured with C curing system for 10 s (0.744 ± 0.014). Where is units for (0.914 ± 0.028) and (0.744 ± 0.014). What about +/- 0.914: is this SED or SE? - Please make all your numbers up to 2 decimal places: (0.744 ± 0.014)........ (0.74 ± 0.01)
- Is there any error bar for Figure 2? How many repeat done?
- Table 4: Where is unit? units are missing
- Table 5: Where is unit? units are missing
Overall, I am happy with the paper but need corrections before accepting this paper fully.
Author Response
Response to Reviewer 1 Comments
Dear reviewer 1,
Thank you for your evaluation and valuable comments.
- Abstract is vague in terms of materials used. The author mentioned that ". For all tests, each material was divided into two groups depending on the curing unit, and each curing unit group was further divided into two subgroups according to curing time. Abstract clearly did not say what is the materials and what is each group. This is not acceptable and need correction.
Response: In the abstract section: We’ve included the materials used, curing units and curing time in parenthesis. The terms ‘group’ and ‘subgroup’ were used for better understanding in the article. Each material was divided into two groups depending on the curing unit (Woodpecker LED-E and CarboLED), and each curing unit group was further divided into two subgroups according to curing time (10 and 20 seconds). There are two groups (group Woodpecker LED-E and group CarboLED) and there are two subgroups (subgroup 10 s and subgroup 20 s).
- Section 3.2 you did not provide units in text for instance look at the following sentence: "The highest mean hardness ratio was recorded
for Filtek Z250 composite resin when cured with C curing system for 10 s (0.914 ± 0.028), while the lowest value was obtained for ACTIVA Bioactive Restorative when cured with C curing system for 10 s (0.744 ± 0.014). Where is units for (0.914 ± 0.028) and (0.744 ± 0.014). What about +/- 0.914: is this SED or SE?
Response: In the section 3.2.: We’ve provided the unit of (VHN of bottom/top) in the text. For example: (0.74 ± 0.01) indicates the mean and standard deviation (SD) hardness ratio (VHN values of bottom/top).
- Please make all your numbers up to 2 decimal places: (0.744 ± 0.014)........ (0.74 ± 0.01).
Response: We’ve made all the numbers up to 2 decimals.
(0.914 ± 0.028) has been replaced by 0.91 ± 0.03.
(0.744 ± 0.014) has been replaced by 0.74 ± 0.01.
- Is there any error bar for Figure 2? How many repeat done?
Response: Figure 2 has been changed and error bars have been added. Please find the attachment for the revised Figure 2.
- Table 4: Where is unit? units are missing
Response: For Table 4, the unit of compressive strength is MPa. It’s stated in the heading of the table. ‘Curing time’ text was added to the table.
- Table 5: Where is unit? units are missing
Response 1: For Table 5, the unit of polymerization shrinkage is %. It’s stated in the heading of the table. ‘Curing time’ text was added to the table.
Response 2: The footnotes of Table 4, the legends of Figure 2 and Table 5 have been revised.
Thank you very much.
Best regards,
Gokcen Deniz BAYRAK

Reviewer 2 Report
Introduction:
- after the hypothesis of the study, mention justification of the study, that is how outcome of teh current study will help clincians.
- Please emphasize on the loop holes in the literatute concerning your study, what is the need of doing this study...i.e., what is yet unknown on this topic.
- what scientific contribution this tudy will add in the literature.
Discussion: focusing on adding strength and limitation of the study in the 2nd last paragraph. add the following refernces to highlit the role of 3D printing and CAD CAM restorations in recent restorative era.
add a few lines about the draw backs of microleakage and its hazard in restorative dentistry, specifically longevity and strength of com;posite.
- Bilgrami A, Alam MK, Qazi FU, Maqsood A, Basha S, Ahmed N, Syed KA, Mustafa M, Shrivastava D, Nagarajappa AK, Srivastava KC. An In-Vitro Evaluation of Microleakage in Resin-Based Restorative Materials at Different Time Intervals. Polymers. 2022 Jan 24;14(3):466.
- Al-Qahtani AS, Tulbah HI, Binhasan M, Abbasi MS, Ahmed N, Shabib S, Farooq I, Aldahian N, Nisar SS, Tanveer SA, Vohra F. Surface Properties of Polymer Resins Fabricated with Subtractive and Additive Manufacturing Techniques. Polymers. 2021 Jan;13(23):4077.
conclusion: remove teh recommendation statement reagridng incremental technique application, if needed add that information in discussion section
Author Response
Response to Reviewer 2 Comments
Dear reviewer 2,
Thank you for your evaluation and valuable comments.
- Introdtction:
- after the hypothesis of the study, mention justification of the study, that is how outcome of teh current study will help clincians.
- Please emphasize on the loop holes in the literatute concerning your study, what is the need of doing this study...i.e., what is yet unknown on this topic.
- what scientific contribution this tudy will add in the literature.
Response:
-In the introduction section,
The last 2 paragraphs have been modified for the fluency of sentences. It was stated that what this study could contribute to the literature. We’ve added 2 more references to the introduction, so the following reference numbers have been changed.
- Discussion: focusing on adding strength and limitation of the study in the 2nd last paragraph. add the following refernces to highlit the role of 3D printing and CAD CAM restorations in recent restorative era.
add a few lines about the draw backs of microleakage and its hazard in restorative dentistry, specifically longevity and strength of com;posite.
- Bilgrami A, Alam MK, Qazi FU, Maqsood A, Basha S, Ahmed N, Syed KA, Mustafa M, Shrivastava D, Nagarajappa AK, Srivastava KC. An In-Vitro Evaluation of Microleakage in Resin-Based Restorative Materials at Different Time Intervals. Polymers. 2022 Jan 24;14(3):466.
- Al-Qahtani AS, Tulbah HI, Binhasan M, Abbasi MS, Ahmed N, Shabib S, Farooq I, Aldahian N, Nisar SS, Tanveer SA, Vohra F. Surface Properties of Polymer Resins Fabricated with Subtractive and Additive Manufacturing Techniques. Polymers. 2021 Jan;13(23):4077.
Response:
-In the discussion section,
We’ve mentioned the restorations prepared with CAD/CAM and 3D printing systems. Also, we’ve added some lines regarding the drawbacks of microleakage and its relation with polymerization shrinkage. The citations for the added sentences were written in the references section. You can find the corrections between lines 434 and 560. We’ve added the references you specified.
- conclusion: remove teh recommendation statement reagridng incremental technique application, if needed add that information in discussion section
Response:
-In the conclusion part,
We’ve removed the recommendation statement and revised this part.
Thank you very much.
Best regards,
Gokcen Deniz BAYRAK